# The Critical Role of Iron in Pregnancy, Puerperium, and Fetal Development

**DOI:** 10.3390/jcm14103482

**Published:** 2025-05-16

**Authors:** Katarzyna Zych-Krekora, Oskar Sylwestrzak, Michał Krekora

**Affiliations:** 1University of Social Sciences, 90-229 Lodz, Poland; 2Department of Perinatology, Obstetrics and Gynecology, Polish Mother’s Memorial Hospital Research Institute in Lodz, 93-338 Lodz, Poland; 3Department of Obstetrics and Gynecology, Polish Mother’s Memorial Hospital Research Institute in Lodz, 93-338 Lodz, Polandkrekoram@poczta.onet.pl (M.K.); 4Department of Prenatal Cardiology, Polish Mother’s Memorial Hospital Research Institute in Lodz, 93-338 Lodz, Poland

**Keywords:** pregnancy, anemia, iron deficiency

## Abstract

**Background/Objectives**: Iron is a fundamental micronutrient. Its deficiency could have a potentially harmful influence on maternal and fetal well-being. **Methods**: This review synthesizes current evidence on the epidemiology, consequences, and clinical meaning of iron deficiency (ID) and iron deficiency anemia (IDA) in pregnant women. **Results**: Untreated ID in pregnancy is associated with a wide spectrum of adverse outcomes: maternal clinical symptoms, cardiovascular disturbances, preterm birth, low birth weight, and impaired fetal neurodevelopment. Furthermore, ID has been related to impaired implantation, miscarriage, congenital heart defects, and neurological complications in fetuses. Women with gastrointestinal disorders and low socioeconomic status constitute a high-risk group of developing IDA. ID remains underdiagnosed and suboptimally managed in some clinical practices. **Conclusions**: This review highlights the critical importance of early detection, individualized supplementation, and public health interventions aimed at reducing iron deficiency during pregnancy.

## 1. Introduction

Although iron has long been known as an essential micronutrient, it is only in the context of pregnancy that it becomes apparent how its role emerges [1]. During pregnancy, the need for iron gradually increases, which is related to both the needs of the mother’s body and the developing fetus. The element plays an important role in hematopoiesis, supports tissue function, and participates in the maturation of key fetal organs [1]. It is important to note that the impact of iron extends far beyond its role in oxygen transport, affecting enzymatic activity, immune function, and cognitive development, thus highlighting its status as a critical nutrient [2]. Iron deficiency can therefore be the initiating factor for various adverse health effects, from anemia and fatigue to cognitive impairment and increased susceptibility to infection.

Pregnancy is, above all, a period of intense biological change, which at the same time represents a particularly sensitive time in a woman’s life. Numerous physiological adaptations on the part of the organism are necessary to ensure optimal conditions for the development of the fetus. With the increase in blood volume, the development of the placenta, and the storage of iron by the fetus, the demand for this element increases [3]. When analyzing the general data, it is important to consider that diet alone does not meet the increased iron requirements during pregnancy, especially under conditions of limited availability of whole foods and insufficient adherence to dietary recommendations [2]. This increased demand, combined with inadequate iron intake, predisposes pregnant women in the long term to iron deficiency (ID), which, if not compensated for, leads to full-blown iron deficiency anemia (IDA), which has become a global problem among pregnant women. This highlights the critical need for effective screening and implementation of further intervention strategies [4].

Due to the multidimensional nature of iron deficiency and its broad consequences—from implantation and fetal development to maternal health and postpartum outcomes—a narrative review format was chosen. This approach offers more freedom to explore the connections between different aspects of the problem and helps bring together data from various fields in a way that is clinically meaningful and coherent.

## 2. Epidemiology

Iron deficiency is the most common nutrient deficiency worldwide. Current epidemiological data are alarming—it is estimated that more than 2 billion people worldwide may be struggling with iron deficiency [5]. Iron deficiency anemia remains a significant health problem in developed countries, affecting approximately 111 million people, or 9.1 percent of the population. Most cases of anemia are due to iron deficiency, but not all anemias are caused by iron deficiency.

Iron deficiency accounts for ≥72% of anemia during pregnancy and 20–37% after delivery [6]. In up to 52% of pregnant women in developing countries, iron deficiency remains a common health problem [7]. This significant prevalence highlights the widespread nature of the problem and the urgent need for targeted interventions. Iron deficiency figures in India remain exceptionally high, with rates of 68.4 percent in children and 66.4 percent in women in 2019, highlighting the severity of the problem in specific regions. The latest official anemia assessment from the fifth National Family Health Survey (NFHS-5) 2019–2021 suggests that despite decades of policy interventions, anemia in India has only worsened [8]. The prevalence increased from 53.2 percent to 57.2 percent in women of reproductive age and from 58.6 percent to 67.1 percent in children compared to the previous NFHS-4 conducted in 2015–2016; this highlights the persistent and ever-present challenge of iron deficiency. In Poland, the prevalence of iron deficiency during pregnancy is estimated at 26.2%, despite the country’s relatively high socio-economic status [9]. The lack of routine diagnostics—including determining ferritin levels—may contribute to underestimating the prevalence of iron deficiency in Poland, highlighting the need for improved screening practices.

## 3. Physiological Distribution of Iron—A Starting Point for Understanding Deficiency

This publication aims to comprehensively analyze iron deficiency’s impact on maternal and child health based on data on body iron distribution versus various aspects of maternal and fetal well-being. By reviewing current clinical studies, this article aims to clarify the multifaceted consequences of iron deficiency during pregnancy and postnatal periods. The consequences of iron deficiency on fetal development, including its potential impact on neurodevelopmental outcomes and long-term health consequences, will be discussed below. Understanding the complexities associated with iron deficiency during pregnancy is essential for health professionals, researchers, and policymakers alike, enabling the development of targeted interventions to mitigate the adverse effects of this common condition.

Although iron deficiency is easy to diagnose and treat, it is often overlooked in pregnancy, and its treatment is suboptimal due to insufficient knowledge of iron supplementation among healthcare professionals [10]. Many clinicians overprescribe oral iron, which has limited benefits as doses increase and can cause gastrointestinal side effects [10].

A prospective cohort study conducted in Ireland in 2024 (McCarthy et al., 2024) observed a significant increase in the prevalence of iron deficiency in first-born women with low-risk pregnancies, from 4.5 percent at week 15 to more than 50 percent at week 33 of pregnancy (for ferritin <30 µg/L), and using a more restrictive ferritin threshold of <15 µg/L, found an increase up to 83.8 percent in the third trimester [11].

In light of these results, it was estimated that ferritin levels below 60 µg/L in the first trimester of pregnancy are a valuable indicator of iron deficiency in the third trimester of pregnancy. Early identification of women at risk of iron deficiency may reduce the risk of adverse maternal and neonatal outcomes [12]. The above study’s findings are supported by the fact that a maternal ferritin level below 13.4 µg/L, a critical threshold, indicates a significant risk of symptomatic and potentially harmful fetal deficiency [3]. Among pregnant women, groups with a higher risk of iron deficiency anemia can be distinguished. These include women with multiple pregnancies due to higher iron requirements for more than one fetus and women with insufficient iron intake, especially those who lack haem iron from meat and fish and non-haem iron from plants. In addition, women with short intervals between pregnancies may not have enough time to rebuild their iron stores before a renewed period of increased demand [13].

Severe vomiting during the first trimester of pregnancy can cause loss of nutrients, including iron. Also, conditions that increase the risk of iron deficiency include chronic inflammatory or malabsorptive diseases such as coeliac disease, Crohn’s disease, or a growing group of women who have undergone bariatric surgery. Finally, it is also important not to forget women of low socioeconomic status, who may be at increased risk of ID due to limited access to iron-rich foods and prenatal care [10]. Identifying the above risk factors is crucial for targeted interventions and improved maternal and child health outcomes [14].

The mother’s iron requirement has been calculated to increase from 18 to 27 milligrams per day during pregnancy, with the most significant increase occurring in the third trimester. In addition, the fetus accumulates 250 milligrams of iron throughout pregnancy, with 80% of this iron being absorbed in the last trimester. Seemingly, the iron requirement only decreases during the first trimester of pregnancy as menstruation ceases, with a median saving of 0.56 mg Fe/d [15].

It is also worth remembering that the increased need for iron also persists during breastfeeding, as breast milk has a relatively low content of this element in its composition [16]. Furthermore, the iron reserves accumulated during fetal development are crucial for the health and well-being of the infant for the first six months of life; this emphasizes the need to ensure sufficient stores of this element stockpiled during pregnancy [10]. After about six months of life, infants’ iron stores become progressively depleted, which, combined with rapid growth, increases the risk of iron deficiency [17].

In iron-deficient pregnant women, the risk of infant anemia at 8–9 months of age increases threefold [18], and maternal iron deficiency during pregnancy may predispose the offspring to develop iron deficiency in infancy, with potentially lifelong sequelae of the deficiency [19].

Optimal fetal development depends on an adequate iron supply, with the fetal brain requiring about 8 mg/kg and red blood cell production requiring a much higher amount of 55 mg/kg. Iron stores in the fetus, mainly in the liver (~12 mg/kg), serve as a key reserve during breastfeeding when the iron content of breast milk is low [20]. When these liver stores are depleted, the brain and other tissues suffer, potentially leading to neurological symptoms, which can manifest when maternal ferritin levels fall below 13.4 µg/L. Maternal anemia significantly increases the risk of low birth weight and fetal iron deficiency, affecting neurodevelopmental outcomes [10,21]. Factors such as hypertension, maternal diabetes, and smoking can interfere with iron transport, further worsening fetal iron status [18]. Given the key role of iron in neurodevelopment, maternal anemia during pregnancy is associated with an increased risk of neurodevelopmental disorders such as autism spectrum disorders, attention-deficit/hyperactivity disorder, and intellectual disability in the offspring [10,22]. Therefore, maintaining optimal maternal iron levels is of paramount importance to protect the health and well-being of both the mother and the developing fetus. Iron is essential for many developmental processes, including myelination and arborisation of dendrites [22].

## 4. Normal Pregnancy Development and Iron Deficiency

### 4.1. Iron Deficiency and Reproductive Outcomes: Implantation, Infertility, Miscarriage, and Ectopic Pregnancy

During the luteal phase of the cycle, the endometrium undergoes intense remodeling to facilitate embryo implantation [23,24]. Iron, a key cofactor for enzymes involved in DNA synthesis, supports the rapid cell division and endometrial maturation necessary for this process. Iron is essential for many developmental processes, including myelination and arborisation of dendrites. Iron deficiency may impair uterine epithelial cell proliferation, potentially leading to a thin endometrium and impeding implantation. In particular, iron deficiency can alter endometrial receptivity [25], the ability of the endometrium to allow the embryo to attach, invade and grow [26]. Therefore, maintaining adequate iron levels is crucial for optimal endometrial function and successful implantation.

Studies indicate a potential link between iron status and unexplained infertility [26]. In particular, women with unexplained infertility were found to have lower transferrin saturation and mean corpuscular hemoglobin concentrations, as well as a higher prevalence of ferritin levels below 30 µg/L compared to fertile controls. Therefore, screening for iron deficiency may be valuable in assessing and supporting women experiencing unexplained infertility [27].

Building on the established links between iron, implantation, and miscarriage risk, it is crucial to highlight the implications for pregnancy outcomes. As mentioned previously, low serum ferritin levels (<30 µg/L) are associated with an increased risk of miscarriage in patients with infertility, and correction of iron stores may improve live birth rates [28]. Maternal anemia can lead to low birth weight and iron deficiency in the fetus, affecting neurodevelopmental outcomes; maintaining optimal maternal iron levels is paramount. This involves early identification of at-risk women and targeted interventions to ensure sufficient iron stores throughout pregnancy [10].

One group particularly vulnerable to iron deficiency is patients qualified for IVF infertility treatment. In one study, these women reported that prior to iron administration in the form of carboxymaltose, 45.5% of them had experienced at least one pregnancy loss. After treatment, this percentage decreased to 20.0%, representing a relative risk reduction of 56.04%. A ferritin level below 30 µg/L doubles the risk of miscarriage in infertile patients while increasing ferritin levels significantly reduces this risk and improves the live birth rate (increase from 36.3% to 58.2%, *p*< 0.001) [28].

Increasing evidence suggests that iron deficiency may be one of the elements that increase the risk of ectopic pregnancy. One proposed mechanism is impaired embryo transport in the fallopian tube, resulting from hypoxia of the smooth muscle (myometrium), which may lead to abnormal development of the endometrium, hindering proper implantation and leading to chronic inflammation that damages the fallopian [29]. It is estimated that iron deficiency may play a role in the pathogenesis of up to 10 percent of ectopic pregnancies, but this relationship remains the subject of further research. Although ectopic pregnancy accounts for only a tiny percentage of all pregnancies, it is associated with a high mortality rate if diagnosis and treatment are delayed [30,31]. At the global level, the burden of maternal disorders due to iron deficiency shows a declining trend in most regions, except in areas where high mortality associated with HIV/AIDS infection persists [29].

### 4.2. Iron Deficiency and the Risk of Congenital Heart Defects (CHD)

Iron deficiency in pregnancy is not only a factor that makes it difficult to maintain—its possible influence on the development of structural heart defects (CHD) in the fetus is increasingly highlighted. Experimental studies have shown that abnormal iron metabolism during embryogenesis can disrupt cardiogenesis, leading to permanent anatomical defects. Among other things, the role of oxidative stress, dysregulation of signaling pathways, and abnormalities in the expression of genes responsible for cardiac development are indicated [32]. However, the most likely mechanism is retinoic acid signaling pathway disruption during the early differentiation of cardiac progenitor cells [32]. In an animal model study, it was shown that in mice fed an iron-deficient diet, in which iron deficiency was confirmed, up to 15% of pregnancies ended in intrauterine death. Among live fetuses, up to 66% of cases had a confirmed heart defect. In comparison, in the control group—which received an iron-rich diet—the percentage of heart defects was only 3%, and no prenatal deaths were observed [32]. Given that congenital heart defects are the most common disability in humans, with a rate of 0.9–1.0% of births, this relationship suggests that iron deficiency may act as an important risk factor for cardiogenesis disorders [23]. Preventing maternal iron deficiency by administering iron early in pregnancy may significantly reduce this risk. A similar study in humans also reported a significant association between maternal iron status and congenital heart defects [33]. In a case-control study including mothers of children with congenital heart defects, 70.7 percent of women were found to be iron deficient, compared to 42.4 percent of mothers of children without defects [33]. Mothers whose fetuses had little CHD were also less likely to consume higher amounts of total and haem iron and less likely to take iron supplements during pregnancy [33].

### 4.3. Neurodevelopmental and Perinatal Effects of Maternal Iron Deficiency

Iron is transported to the fetus via the transferrin receptor in the placenta. However, in iron-deficient conditions, the maternal body prioritizes iron for its own needs, limiting its availability to the fetus. This reduced iron transport may interfere with energy metabolism in the fetal brain, as iron is a key cofactor for mitochondrial enzymes responsible for ATP (adenosine triphosphate) production [34]. Consequently, reduced energy production impairs neuronal function. Furthermore, maternal anemia significantly increases the risk of iron deficiency in the fetus, which may impede myelination and dendritic network development in the developing brain [35]. Severe iron deficiency during pregnancy has also been shown to increase fetal morbidity [10]. Therefore, ensuring sufficient iron levels during pregnancy promotes optimal neurodevelopment and prevents long-term cognitive deficits [35]. Scientific reports highlight that prenatal iron deficiency affects the expression of genes related to neuronal differentiation, synaptic plasticity, and neuroprotection. This may affect the mTOR (Mammalian Target of Rapamycin) pathway and BDNF (brain-derived neurotrophic factor) expression, potentially causing long-term brain dysfunction that persists even after iron deficiency is corrected after birth [36]. Consequently, preschool and school-age children may show deficits in attention, working memory, and language skills, significantly affecting their academic performance.

Given the potential link between maternal iron intake and the child’s risk of developing autism spectrum disorders (ASD), a thorough history of iron supplementation use before, during, and after pregnancy and lactation is particularly important [37]. In a study conducted as part of the CHARGE project, 520 mothers of children on the autism spectrum and 346 mothers of typically developing children participated. They analyzed iron intake from supplements, multivitamins, and breakfast cereals (rich in iron) at different times: 3 months before pregnancy, during pregnancy, and lactation. Based on existing studies, it is clear that maternal iron deficiency during pregnancy can significantly increase the risk of autism spectrum disorders in children [37]. The CHARGE study showed that pregnant women with lower iron intake were more likely to have children diagnosed with autism spectrum disorder (ASD) [37]. Specifically, mothers who took fewer iron-specific supplements during a key period—which lasted from three months before pregnancy to the end of breastfeeding—were more likely to have a child with ASD [37]. Additionally, when low iron intake was compared and contrasted with advanced maternal age (35 years or older) or metabolic disorders such as obesity, diabetes, and hypertension, the risk of having a child with ASD increased more than fivefold [37]. This suggests that iron plays a key role in early neurodevelopmental processes, and that iron deficiency may interfere with these processes, leading to increased susceptibility to ASD.Nevertheless, more studies are needed. It is worth noting that iron deficiency affects a significant proportion of pregnancies, between 40% and 50% [38]. Given these findings, ensuring adequate iron intake through supplementation and dietary modifications is crucial for pregnant women to reduce the risk of ASD in their offspring [37]. Further research is needed to fully elucidate the mechanisms of this relationship and develop targeted prevention strategies [37]. Building on previous findings, maternal anemia during pregnancy is further complicated by several additional risk factors [22]. Late anemia is more common in mothers with obesity, a positive psychiatric history, and who are of older age (over 40 years) [22]. Early anemia, diagnosed before the 30th week of pregnancy, is more common in mothers with lower education, lower income, and who are underweight [22]. Babies of mothers with early anemia are more likely to be born prematurely and small for gestational age, while late anemia is associated with higher birth weight [22]. A study of 532,232 children born to 299,768 mothers in Sweden between 1987 and 2010 found that early anemia was associated with an increased risk of neurodevelopmental disorders in children [22]. Specifically, early anemia was associated with a 44% increased risk of autism spectrum disorders, a 37% increased risk of ADHD, and a 120% increased risk of intellectual disability [22]. Continuing this line of research, another project conducted at a university hospital in Finland analyzed a group of 215 pregnant women with anemia (Hb <10.0 g/dL).

One study reported a clearly higher rate of perinatal complications compared to a control group of 11,545 pregnant women without anemia who delivered in the same center and during the same period [39]. The iron-deficient group had a higher risk of preterm birth (10.2% vs. 6.1% in the control group; *p* = 0.009) and a higher incidence of fetal growth restriction (1.9% vs. 0.3% in the control group; *p* = 0.006) [39]. Additionally, there was a significantly higher rate of postpartum infections (9.3% vs. 3.5%; OR 2.86; 95% CI: 1.79–4.59; *p*< 0.001) and a longer average postpartum hospital stay by one day in women with anemia (*p*< 0.001) [39].

Women with anemia also required postpartum blood transfusions significantly more often than those without anemia (5.6% vs. 2.6%; OR 2.48; 95% CI: 1.37–4.49; *p* = 0.002) [39]. Although the incidence of massive postpartum hemorrhage (defined as blood loss >1000 mL) did not differ significantly between groups, anemic women were more susceptible to uterine atony—a leading cause of postpartum hemorrhage—and delayed uterine involution, which may lead to prolonged bleeding and increased infection risk. These disturbances, together with chronic tissue hypoxia and impaired immune response, may contribute to a higher rate of severe maternal morbidity, including sepsis, thromboembolic events, and the need for intensive care unit admission [39].

These findings highlight the broader consequences of iron deficiency anemia in pregnancy—extending beyond neurodevelopmental risk—including serious perinatal complications, an increased burden on healthcare systems, and potential long-term health risks for both mother and child [10,39].

Given that iron deficiency affects a significant proportion of pregnancies [10], these data highlight the critical importance of proactive screening and managing iron status throughout pregnancy to mitigate this risk and improve maternal and neonatal outcomes (Figure 1).

### 4.4. Restless Legs Syndrome and Iron-Related Neurological Changes in Pregnancy

Iron deficiency contributes significantly to restless legs syndrome (RLS) during pregnancy, affecting between 20% and 30% of pregnant women [40]. RLS symptoms usually appear in the second trimester of pregnancy and worsen in the third, often resolving within a few weeks after delivery (in 70–90% of women) [41]. RLS increases the risk of depression, insomnia, and stress, and in severe cases, can be associated with preterm birth [42]. Women who experience RLS during one pregnancy have a 30% chance of recurrence in subsequent pregnancies [41]. Given that iron deficiency is a known risk factor for RLS [41], adjusting iron levels through diet or supplementation may help alleviate symptoms [43]. One likely mechanism for the onset of RLS is iron deficiency in the central nervous system. The increased need for iron during pregnancy may limit its availability to the brain, which promotes increased oxidative stress and disrupts nerve conduction, thereby increasing the risk of restless legs syndrome (RLS) [18].

### 4.5. Cardiac Function and ECG Changes in Pregnant Women with Iron Deficiency

Iron deficiency during pregnancy has far-reaching consequences for the mother-to-be beyond the commonly recognized symptoms of fatigue and irritability [10]. Emerging research has highlighted the impact of iron deficiency on heart function in pregnant women, and studies comparing electrocardiographic parameters between iron-deficient and healthy pregnant women have shown significant differences. One such study, conducted in India, assessed the effect of iron deficiency anemia on myocardial function using an ECG during the second trimester of pregnancy [44]. The study reported a significant shortening of the QRS complex duration, a prolongation of the QTc interval in the iron deficiency group, and more frequent T-wave abnormalities [44]. Notably, 90% of individuals in the study group showed tachycardia and ECG abnormalities, with a negative correlation observed between hemoglobin, serum ferritin, and these cardiac abnormalities (Table 1) [44]. These findings suggest that iron deficiency anemia in pregnancy may induce ECG changes, potentially leading to cardiac hypertrophy due to anemia and volume overload [44].

Furthermore, maternal iron deficiency can result in breathing difficulties, syncope, palpitations, or sleep disturbances [7]. It is also associated with an increased risk of perinatal infection, pre-eclampsia, and bleeding, as well as postpartum cognitive impairment and behavioral difficulties [7].

### 4.6. Immune Suppression and Maternal Outcomes in Iron Deficiency Anemia

Iron deficiency anemia in pregnant women can also impair immune function and increase susceptibility to infectious diseases [46]. Studies have shown that pregnant women with IDA have significantly lower T-lymphocyte subsets (CD3 + and CD4 + cells), serum interleukin-2, and IgG (immunoglobulin G) levels than healthy women without IDA [46]. The incidence of infectious diseases is significantly higher in pregnant women with iron deficiency anemia (IDA) compared to pregnant women without IDA [46]. This highlights the significant impact of IDA on cellular immune function and the risk of infectious diseases during pregnancy [47]. The CD4/CD8 ratio indicates a balance between helper T cell responses and cytotoxic T cells. A value above 1 indicates CD4+ T-cell dominance, while a value below 1 may suggest immunodeficiency [48]. The CD4⁺/CD8⁺ lymphocyte ratio in healthy individuals is usually between 1.5 and 2.5, reflecting a normal balance between helper and cytotoxic lymphocytes. In pregnant women with iron deficiency anemia (IDA), this value was reduced to about 1.2, which may indicate a disturbed immune response and reduced cellular immunity [48].

Pregnant women with IDA had significantly more infections than healthy women, confirming the effect of anemia on immunosuppression (*p* < 0.05). The infection rate is highest in pregnant women with iron deficiency anemia, reaching about 40%. The infection rate in healthy pregnant women is around 15%, indicating a significantly lower risk of infection compared to women with IDA. The lowest infection rate is observed in women who are not pregnant, around 10%. Therefore, pregnancy with iron deficiency anemia is associated with a significantly increased risk of infection compared to both healthy pregnant women and nonpregnant women. These findings highlight the importance of monitoring and treating IDA in pregnant women to reduce the risk of infection. Based on an extensive study in California involving 3.86 million births, anemia during pregnancy is a significant risk factor for severe maternal morbidity [10]. The study found that 12.6 percent of women had anemia, and the prevalence of anemia increased 2–3-fold between 2011 and 2020 across all ethnic groups. Anaemia was associated with a 3.5-fold increased risk of severe pre-eclampsia, a 3.2-fold increased risk of postpartum hemorrhage requiring transfusion, a 2.8-fold increased risk of acute kidney injury, a 2.5-fold increased risk of preterm birth and a 2.3-fold increased risk of low birth weight [10,49]. Overall, 1 in 5 cases of major maternal complications can be attributed to anemia [10]. These findings highlight the critical importance of monitoring and treating anemia during pregnancy to reduce the risk of severe maternal morbidity and improve maternal and neonatal outcomes. In addition, there are racial and ethnic differences in iron deficiency and iron deficiency anemia [10].

### 4.7. Perinatal Mental Health and Iron Status

Another important aspect of iron deficiency during pregnancy is its association with maternal depression [45]. A systematic review and meta-analysis of 15 studies involving 32,792,378 women found a statistically significant association between anemia and maternal depression. The review found that perinatal depression affects approximately 16% of women during pregnancy and 12% of women after delivery, with even higher rates in low-income countries (25.3% during pregnancy and 19% after delivery) [50]. Women with anemia had a 36% increased risk of depression during pregnancy and a 53% increased risk of postpartum depression [45]. Proposed mechanisms linking anemia to depression include the important role of iron in brain function, particularly in the synthesis of serotonin and dopamine [45]. Iron deficiency can lead to reduced neurotransmitter synthesis, hormonal imbalances, and reduced brain oxygenation, all of which can negatively impact mood and cognitive function [45,50].

### 4.8. Impact of Iron Deficiency on Breastfeeding

Iron deficiency can significantly affect breastfeeding, affecting milk production and quality [51,52]. Studies have shown that anemic mothers produce 20–25% less milk than mothers without anemia. Furthermore, the iron concentration in the milk of anemic mothers is reduced by 50% [10]. Consequently, the average breastfeeding time is shorter in anemic mothers, approximately 4 months, compared to 6 months in non-anemic mothers [10]. These findings highlight the importance of screening and treating anemia in the postnatal period to support maternal health and optimal infant nutrition [10] Reducing iron levels through diet or supplementation can help alleviate these issues and ensure adequate infant milk supply and iron content [51].

In Poland, while the initial percentage of women initiating breastfeeding after delivery is high, it decreases significantly over time [53]. A 2015 report showed that 91.5 percent of pregnant women expressed an intention to breastfeed, and 81.6 percent of mothers exclusively breastfed after discharge from the hospital. However, by the child’s sixth month, this rate had dropped to 35.2% [53]. The most significant decline in breastfeeding rates in Poland occurs around the infant’s 2nd to third month of life [53]. Therefore, early detection and treatment of anemia can significantly reduce the risk of maternal depression, and treatment of iron deficiency should become standard in both prenatal and postnatal care [45].

In summary, iron deficiency during pregnancy is an often underestimated but critical deficiency with far-reaching consequences for both the mother and the developing baby [10,18]. Its impact includes increasing the risk of severe maternal illnesses such as pre-eclampsia and postpartum hemorrhage, as well as contributing to maternal depression, impaired immune function, and cardiac abnormalities [10]. In particular, iron deficiency anemia can induce ECG changes, potentially leading to cardiac hypertrophy due to anemia and volume overload. In addition, iron deficiency can significantly affect fetal development, leading to low birth weight, neurological disorders, and long-term health problems [10,38]. The effects of breastfeeding, including reduced milk production and lower iron content in breast milk, further emphasize the importance of managing iron deficiency in the postnatal period [10]. Given the high prevalence of iron deficiency among pregnant women—affecting more than 40% of pregnancies in the first trimester of pregnancy in the U.S. [3,10]—and its wide range of adverse effects, routine screening and appropriate iron supplementation tailored to individual iron status is essential to ensure the well-being of both mother and child. Addressing this deficiency should be a priority in prenatal and postnatal care to mitigate its profound and lasting effects [10]. Public health strategies should educate the population about the need for a healthy diet and iron supplementation before conception or at least early in pregnancy [7].

## Figures and Tables

**Figure 1 jcm-14-03482-f001:**
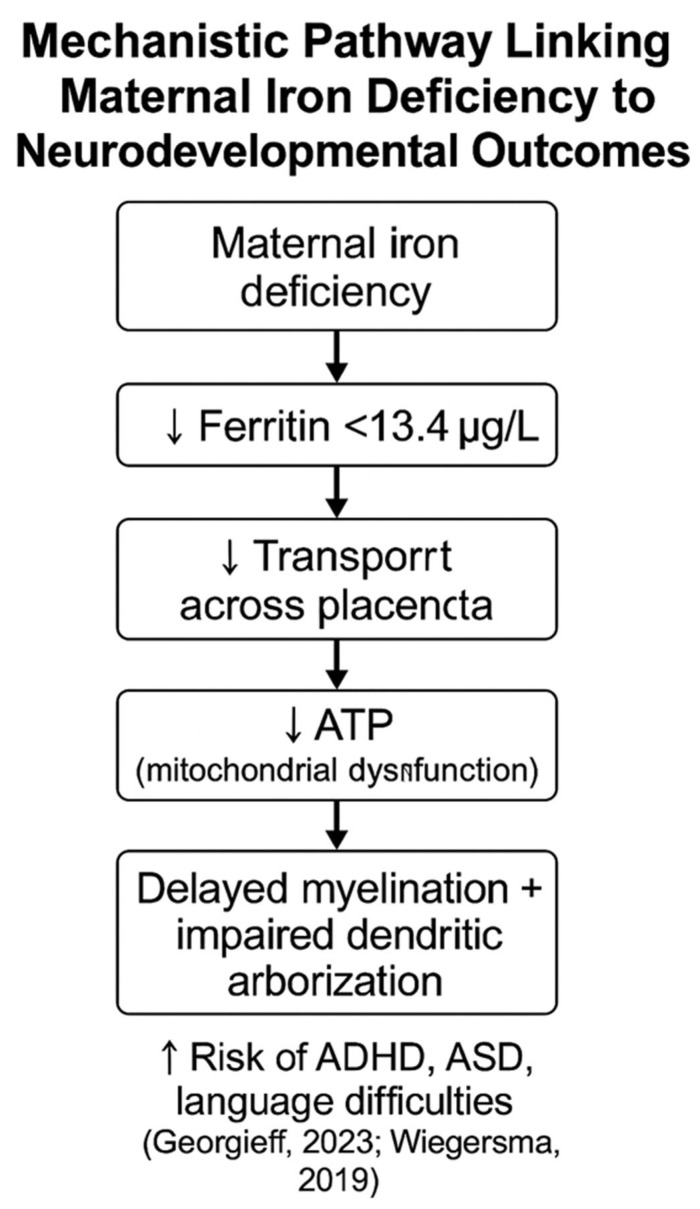
Pathway of potential link between maternal iron deficiency and neurodevelopmental fetal outcome [20,22].

**Table 1 jcm-14-03482-t001:** Key health consequences of iron deficiency during pregnancy and postpartum.

Health Domain	Impact of Iron Deficiency	Source
Mother	Anemia and fatigue	Benson et al., 2022 [10]
Mother	Cardiac rhythm disturbances (QTc, tachycardia)	Sifakis and Pharmakides, 2000 [44]
Mother	Restless legs syndrome (RLS)	Srivanitchapoom et al., 2014 [41]
Mother	Prenatal and postpartum depression	Goshtasebi et al., 2013 [45]
Mother	Impaired immunity, increased risk of infections	Sobhani et al., 2011 [46]
Mother	Lactation difficulties—reduced milk volume and iron content	Benson et al., 2022 [10]
Mother	Increased risk of perinatal complications (hemorrhage, infections, prolonged hospitalization)	Kemppinen et al., 2020 [39]
Mother	Higher risk of cardiovascular complications (cardiac hypertrophy)	Sifakis and Pharmakides, 2000 [44]
Fetus/ Newborn	Low birth weight	Alwan et al., 2015 [21]
Fetus/ Newborn	Fetal growth restriction (FGR)	Kemppinen et al., 2020 [39]
Fetus/ Newborn	Increased risk of congenital heart defects	Kalisch-Smith et al., 2021 [32]
Fetus/ Newborn	Neurodevelopmental disorders (e.g., autism, ADHD, intellectual disability)	Wiegersma et al., 2019 [22]
Fetus/ Newborn	Disrupted myelination and dendritic development	Georgieff, 2023 [20]
Fetus/ Newborn	Higher risk of miscarriage and implantation failure	Silfvast and Simberg, 2022 [28]
Fetus/ Newborn	Increased risk of ectopic pregnancy	Wu et al., 2024 [29]
Fetus/ Newborn	Impaired energy metabolism in the fetal brain	Zaugg et al., 2022 [34]
Infant	Risk of anemia by 9 months of age	Georgieff, 2020 [18]
Infant	Attention, working memory and language deficits	Özyurt and Bulutlar, 2024 [36]
Infant	Shortened breastfeeding duration	Benson et al., 2022 [10]

## Data Availability

Data available after reasonable request.

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
