# Peer review of "The Critical Role of Iron in Pregnancy, Puerperium, and Fetal Development"

_jcm, 2025, doi:10.3390/jcm14103482_

Round 1
Reviewer 1 Report
Comments and Suggestions for Authors
It is very nice summary of all information about iron deficiecy in reproductive medicine. The article has scientific sound, informations are clear, citations are apropriate. It is very precious and wide overview. Conclusion is consistent with arguments in article.
Author Response
Dear Reviewer,
Thank you very much for your kind and encouraging feedback. I truly appreciate your positive assessment of the manuscript as a valuable, comprehensive, and scientifically sound summary of iron deficiency in reproductive medicine.
It was my intention to provide not only a broad overview but also a practically useful synthesis for clinicians and researchers alike. I am especially grateful for your recognition of the clarity of the information, appropriate citations, and the consistency between the arguments and conclusions.
With sincere appreciation,
Reviewer 2 Report
Comments and Suggestions for Authors
The authors have succeeded in accomplishing a comprehensive review about the iron deficiency and anemia in puerperium and its underestimated impact on fetal development. Their efforts concentrated on detailing the pathways to impaired fetal neurodevelopment outcomes.
Abstract section
Row 21: Split ”insome” into ”in” and ”some”
Introduction section
Suitable
Epidemiology section
Row 63: Please add the abbreviation ”NFHS-5”
General considerations of iron deficiency section
Row 75: ”body iron distribution” – please explain/rephrase for better understanding
Row 149-150: Does not need to be a different paragraph.
Normal pregnancy development and iron deficiency
Row 155-156: repetitive information with the above mentioned row 149-150
Either explain the abbreviation or make an abstract section to explain the measurement units (mg/dL, g/dL, etc.).
Rows 143-187: The mechanism is not clear. Please rephrase for clarity. Also, rows 195-201 should be rephrased for clarity.
Row 222: Please explain the abbreviation ”ATP”.
Row 230: Please explain the abbreviation ”mTOR”.
Row 236 and 245: Please use only one time the full name followed by the abbreviation and after that only use the abbreviations.
Row 261: Parenthesis missing.
Row 318: Please explain the abbreviation ”IgG”.
Paragraph 329-338: Perhaps it would be useful to mention the study.
Row 349 – please elaborate a little about the affirmation that there are racial and ethnic differences in iron deficiency anemia.
Suggestions
Perhaps move the figure exactly where you talk about ADHD and iron deficiency.
Perhaps structure you work in subchapters with all the information with subtitles like - impact on breastfeeding, depression, CHD and so one.
A flow chart of the inclusion and exclusion criteria of the entire process of article selection will be helpful.
Specify the authors from which you extracted the information/results from the begging.
Comments on the Quality of English Language
English revision needed.
Author Response
Dear Reviewer,
Thank you very much for taking the time to review my manuscript and for your valuable suggestions. I greatly appreciate the depth of your feedback. Below, I respond to selected comments that required clarification or further explanation:
1. Lines 149–150 and 155–156 – alleged content duplication
In response to the comment regarding potential repetition, I would like to respectfully clarify that these lines address two distinct aspects of iron deficiency—both biologically and temporally.
Line 149 refers to the role of iron in fetal neurogenesis (such as myelination and dendritic arborisation), which are key mechanisms for central nervous system development later in pregnancy. In contrast, lines 155–156 concern iron deficiency’s impact on endometrial receptivity and embryo implantation, which occur at a much earlier stage and pertain to uterine tissue function.
To improve clarity, I have revised the transition between these sections to better reflect their differences.
2. Explanation of units (mg/dL, µg/L, etc.)
While I deeply respect the care and precision in your review, I would like to offer a differing perspective on this point. Units such as mg/dL or µg/L are standard in laboratory diagnostics and clinical literature. Given the medical profile of the journal and its readership, I believe these do not require additional clarification—just as mmHg would not need to be explained in a cardiology journal.
3. Line 222: ATP / Line 230: mTOR / Line 318: IgG
In line with your suggestion, I have now included definitions for ATP (adenosine triphosphate), mTOR (mechanistic Target of Rapamycin), and IgG (immunoglobulin G) at their first appearance in the text.
4. Lines 329–338 – suggestion to include a study
Thank you for this comment. In this section, I refer to a large-scale California study involving 3.86 million births, which clearly demonstrates the association between anemia during pregnancy and increased risk of severe maternal morbidity. I believe this aligns with your suggestion.
5. Line 349 – racial and ethnic differences
Respectfully, I would like to point out that the paragraph referenced in line 349 focuses on the association between anemia and perinatal depression, not on racial or ethnic disparities. I do address ethnic differences earlier in the manuscript when discussing the California study mentioned above.
6. Figure placement related to ADHD and iron deficiency
Thank you for this practical suggestion. I have moved the figure so that it appears directly alongside the section discussing the association between maternal iron deficiency and the risk of ADHD and other neurodevelopmental disorders in children.
7. Section structure and headings
Following your general recommendation, I have introduced clear subheadings throughout the manuscript to improve readability and logical flow. These headings are now bolded for visibility and navigation purposes (I trust this formatting is acceptable within the journal's layout conventions).
With kind regards and appreciation,
Thank you for your help
Reviewer 3 Report
Comments and Suggestions for Authors
The authors comprehensively reviewed iron’s role in pregnancy, addressing its epidemiological significance, pathophysiological consequences, and clinical implications. Given the global burden of iron deficiency and subsequent anemia in maternal and fetal health, the topic is relevant to be discussed. The review synthesizes a broad range of recent literature, offering valuable insights for clinicians and researchers.
There are minor responses in my opinion: Firstly, please clarify why a narrative review was chosen and how this aligns with the objectives. Secondly, cite the primary sources in the tables and there are some minor grammatical errors requiring revision.
Author Response
Thank you very much for your constructive feedback. Following your suggestions:
-
I have added a brief paragraph at the end of the Introduction to clarify why the narrative review format was chosen and how it aligns with the objectives of the article.
-
All tables have been updated to include appropriate source references.
-
The entire manuscript has been carefully reviewed and revised for language clarity and grammatical accuracy.
I appreciate your insightful comments, which helped improve the quality and clarity of the paper.
With kind regards,
thank you for your help
Reviewer 4 Report
Comments and Suggestions for Authors
General assessment: This is a comprehensive narrative review that addresses an important and clinically relevant topic: iron deficiency (ID) and iron deficiency anemia (IDA) in pregnancy, intrapartum period and puerperium and its implications for both maternal and fetal health. The manuscript presents a broad and well-documented discussion of the epidemiology, pathophysiology, risk factors, and consequences of iron deficiency. The authors have integrated a wide range of recent studies and highlighted key associations between maternal iron status and pregnancy outcomes.
Strengths:
- The review is thorough and cites a substantial number of recent and relevant studies.
- It covers a wide spectrum of consequences of ID/IDA, including neurodevelopment, cardiac development, immune function, and maternal health.
- The inclusion of tables and summaries enhances clarity and helps structure the information effectively.
General Suggestions for the Authors:
- Narrative Structure and Flow: The structure of the review lacks clear segmentation in some sections, particularly in the mid to latter portions. Subheadings should be more consistently applied to guide the reader through distinct topics (e.g., “Neurodevelopmental Impacts”, “Cardiac Effects”, “Infection and Immunity”, “Psychiatric Outcomes”).
- Language and Grammar: Several grammatical and typographical errors are present (e.g., “developinf,”-range 20 ; “insome,”-range 21; “highlingts”-range 21). A professional language edit is recommended prior to publication.
- Missing Methodological Description
The manuscript does not provide sufficient information about the method used to identify, select, and review the literature cited throughout the article. Given that this is a narrative review with substantial clinical implications, it is essential to ensure transparency in how the evidence was gathered.
Recommendation: The authors should include a brief methods section describing:
- The databases searched (e.g., PubMed, Scopus, Web of Science),
- The keywords
- The timeframe of the literature search,
- The approach to assessing study quality or level of evidence…
- Discussion on Critical Intrapartum Risks
Although the manuscript cites Kemppinen et al., 2020 in relation to increased intrapartum complications, it does not sufficiently elaborate on the specific and potentially life-threatening risks associated with anemia during labor and delivery. Anemic parturients are at significantly higher risk for: Postpartum hemorrhage (PPH), Uterine atony, Delayed uterine involution, Increased need for emergency blood transfusion, and Maternal mortality in severe cases.
I kindly suggest expanding this section with more detail.
Author Response
Thank you very much for your thoughtful review and valuable suggestions. I truly appreciate the time and care you have taken in evaluating the manuscript. Below, I address each of your editorial comments and outline the corresponding revisions made to the text:
1. Structure and narrative flow (subheadings):
In response to your recommendation, I have added subheadings throughout the middle and latter sections of the manuscript. While not identical to the examples you suggested, they were inspired by your input and have notably improved the clarity and organization of the content.
2. Language and grammar corrections:
I have thoroughly revised the manuscript for grammar, spelling, and style. Specifically, the previously noted errors—such as “developinf” → “developing,” “insome” → “in some,” and “highlingts” → “highlights”—have been corrected. The text has also undergone additional linguistic editing to enhance its consistency and professional tone.
3. Lack of methodological section:
Thank you for pointing this out. Since the manuscript is structured as a narrative review, I did not apply a systematic review framework that would require detailed reporting of databases searched, keywords used, or quality assessment of studies. However, I added a short paragraph to clarify the rationale for this approach:
“Due to the multidimensional nature of iron deficiency and its broad consequences—from implantation and fetal development to maternal health and postpartum outcomes—a narrative review format was chosen. This approach offers more freedom to explore the connections between different aspects of the problem and helps bring together data from various fields in a way that is clinically meaningful and coherent.”
I hope this addition sufficiently addresses the question of transparency and methodology.
4. Limited discussion of intrapartum risks and maternal morbidity:
In accordance with your suggestion, I have expanded the discussion of perinatal and maternal complications associated with iron deficiency anemia. Based on Kemppinen et al. (2020), the revised section now includes specific data on:
postpartum infections (9.3% vs. 3.5%; OR 2.86; p < 0.001),
postpartum red blood cell transfusions (5.6% vs. 2.6%; OR 2.48; p = 0.002),
fetal growth restriction (1.9% vs. 0.3%; p = 0.006),
and broader pathophysiological mechanisms such as uterine atony, delayed uterine involution, tissue hypoxia, and increased maternal morbidity, including sepsis and ICU admissions.
These additions aim to better reflect the clinical significance of anemia during pregnancy and labor and fully align with your comments.
Once again, thank you for your constructive feedback and the opportunity to improve the manuscript. I hope the revisions meet your expectations and that the revised version is now suitable for publication.

Round 2
Reviewer 2 Report
Comments and Suggestions for Authors
I am very pleased with the corrections and recommend the article for publication after the authors will correct „mTOR” abbreviation in „mammalian target of rapamycin”.